# The Influence of Baseline Clinical Status and Surgical Strategy on Early Good to Excellent Result in Spinal Lumbar Arthrodesis: A Machine Learning Approach

**DOI:** 10.3390/jpm11121377

**Published:** 2021-12-16

**Authors:** Pedro Berjano, Francesco Langella, Luca Ventriglia, Domenico Compagnone, Paolo Barletta, David Huber, Francesca Mangili, Ginevra Licandro, Fabio Galbusera, Andrea Cina, Tito Bassani, Claudio Lamartina, Laura Scaramuzzo, Roberto Bassani, Marco Brayda-Bruno, Jorge Hugo Villafañe, Lorenzo Monti, Laura Azzimonti

**Affiliations:** 1IRCCS Istituto Ortopedico Galeazzi, 20161 Milan, Italy; pberjano@gmail.com (P.B.); compagnone.nico@gmail.com (D.C.); paolo.barletta@grupposandonato.it (P.B.); fabio.galbusera@grupposandonato.it (F.G.); andrea.cina@grupposandonato.it (A.C.); tito.bassani@grupposandonato.it (T.B.); c.lamartina@chirurgiavertebrale.net (C.L.); scaramuzzolaura@gmail.com (L.S.); r.bassani.spine@gmail.com (R.B.); marco.brayda@spinecaregroup.it (M.B.-B.); 2Istituto Dalle Molle di Studi sull’Intelligenza Artificiale (IDSIA), USI-SUPSI, 6900 Lugano, Switzerland; luca.ventriglia@idsia.ch (L.V.); david.huber@idsia.ch (D.H.); francesca.mangili@idsia.ch (F.M.); ginevra.licandro@idsia.ch (G.L.); laura.azzimonti@idsia.ch (L.A.); 3IRCCS Fondazione Don Carlo Gnocchi, 20161 Milan, Italy; mail@villafane.it; 4Orthopedics and Traumatology Unit, Istituto Clinico Villa Aprica, 22100 Como, Italy; lorenzomonti@hotmail.it

**Keywords:** artificial intelligence, lumbar fusion, degenerative disc disease, adult spine deformity, scoliosis, spine registry, personalized medicine

## Abstract

The study aims to create a preoperative model from baseline demographic and health-related quality of life scores (HRQOL) to predict a good to excellent early clinical outcome using a machine learning (ML) approach. A single spine surgery center retrospective review of prospectively collected data from January 2016 to December 2020 from the institutional registry (SpineREG) was performed. The inclusion criteria were age ≥ 18 years, both sexes, lumbar arthrodesis procedure, a complete follow up assessment (Oswestry Disability Index—ODI, SF-36 and COMI back) and the capability to read and understand the Italian language. A delta of improvement of the ODI higher than 12.7/100 was considered a “good early outcome”. A combined target model of ODI (Δ ≥ 12.7/100), SF-36 PCS (Δ ≥ 6/100) and COMI back (Δ ≥ 2.2/10) was considered an “excellent early outcome”. The performance of the ML models was evaluated in terms of sensitivity, i.e., True Positive Rate (TPR), specificity, i.e., True Negative Rate (TNR), accuracy and area under the receiver operating characteristic curve (AUC ROC). A total of 1243 patients were included in this study. The model for predicting ODI at 6 months’ follow up showed a good balance between sensitivity (74.3%) and specificity (79.4%), while providing a good accuracy (75.8%) with ROC AUC = 0.842. The combined target model showed a sensitivity of 74.2% and specificity of 71.8%, with an accuracy of 72.8%, and an ROC AUC = 0.808. The results of our study suggest that a machine learning approach showed high performance in predicting early good to excellent clinical results.

## 1. Introduction

Degenerative spine disorders represent a complex condition that mainly affects the elderly population, with an incidence in healthy people aged over 70 years of up to 68% [1].

Pain and disability represent its main features, leading to a significant clinical and socio-economic impact with an increasing role in daily medical practice. Its dissemination goes hand in hand with the aging of the population of developed countries. The spinal disorders have a broad spectrum of clinical manifestations: from minimal or asymptomatic to an invalidating condition. The presentation pattern can variably affect segmental, regional, and global alignment. The pain and disability represent the main feature in a way that is comparable with other self-reported chronic conditions in the general population such as congestive heart failure, arthritis, chronic lung disease or diabetes [2].

The therapeutic approach of spinal disorders is challenging in terms of decision making for several causes and symptoms. Furthermore, the decisional process is made even more complicated by aging patients eligible for surgery and different clinical conditions and comorbidities [1].

In the last decades, the rate of spine surgery increased by up to 40%, and several randomized trials demonstrated the positive and significant effects of these procedures [3,4]. Its safety and effectiveness vary widely among patients. In the worse scenarios, the complications rate can be up to 13% [5]. Indeed, there is always room for improvements in terms of the clinical, surgical, and economic points of view [6]. The scientific research in this field targeted to evaluate the improvement in quality of life (QoL) after surgical treatment for spine surgery in relation to patient age, comorbidity and baseline status. With the aim to improve the cost–effect ratio’s performance, the rise of predictive models (PM) is continuously increasing. In 2015, McGirt et al. [7] presented a PM for the clinical practice to help patients, providers, and hospital systems. It is based on demographics, patients’ reported outcomes, and clinical data. In particular, the baseline patient-specific factors, such as symptom duration, smoking status, preoperative comorbidities and mental and physical conditions, seem to significantly influence outcomes following lumbar surgery. Sinikallio et al. [8], in a prospective analysis, demonstrated that the patients with preoperative depression and those who had continuous depression postoperatively experienced poor post-operative surgical outcomes and can benefit from targeted cognitive behavioral therapy [9]. Patient-specific factors beyond medical comorbidities, surgical indications, and surgical approaches can play a significant role in influencing overall patient outcomes [10]. The impact of lumbar spine surgery on patients’ life is commonly evaluated with three patient-reported outcome measures (PROMs): The Oswestry Disability Index (ODI), the Physical Component Score of the Short Form of the Medical Outcomes Study (SF-36 PCS), and pain scales (VAS leg and back). The minimum clinically important difference (MCID) is commonly considered the threshold to measure the effect of the surgery for the single questionnaire. The use of PROMs and their prediction through machine learning approaches represent a milestone in the development of shared, informed, and individualized decision making potentially capable to support the surgeon to choose the right intervention, at the right time, for the right patient [7].

Our study aimed to develop a preoperative machine learning (ML) model to predict a good to excellent early clinical outcome by using baseline demographic and health-related quality of life scores (HRQOL).

## 2. Materials and Methods

### 2.1. Clinical and Demographic Data

The study was conducted in a single spine surgery center and it is based on a retrospective review of prospectively collected data from the institutional registry—SpineReg [11]. The inclusion criteria were age ≥ 18 years, both genders, lumbar arthrodesis procedure identified using the ICD-9 code (8106, 8107 or 8108), a follow up assessment (ODI, SF-36 and core outcome measures index—COMI back) and the capability to read and understand the Italian language. A full set of peri-operative and post-operative data along with clinical outcomes from January 2016 to December 2020 were evaluated. Exclusion criteria were a weak degree of baseline disability or pain (ODI < 20/100 and COMI back < 3/10), number of levels fused not specified and subject not stratified according to Glassman classification.

The study protocol was conducted in accordance with the Helsinki Declaration of 1957 as revised in 2000. The procedures followed the ethical standards of the responsible committee on human experimentation and was approved by the ethics committee of our institution (second amendment to the SPINEREG protocol 14 issued on 13 April 2016). The project was supported with funds from the Italian Ministry of Health (project code CO-2016-02364645). All patients gave their written informed consent for the participation in the registry. Baseline demographics, BMI, gender, comorbidities collected through the comorbidity Charlson index (CCI) [12], diagnosis according to the Glassman classification [13], number of spinal levels of intervention, spinal level indexed surgery, clinical scores resulting from medical surveys, complications and revision surgeries were collected.

Table 1 shows that the rates of missing data ranged from 0.0% for the baseline scores of the PROMs and for the patient’s personal information, to 20.4% for Levels variables. Independent variables with at least one missing value were imputed using predictive mean matching for numerical variables, and binary/multinomial logistic regression or an ordered logit model for categorical variables. Outcome variables were not imputed to avoid the introduction of bias into the results. Only patients with the observed outcome variables were included in the analysis.

### 2.2. Clinical Outcomes

The primary outcome was the early (six months post op) significant clinical improvement. In particular, value of improvement higher than 12.7 for ODI [14], 6 for SF36-PCS and 2.2 for COMI Back were considered as indicators of significant clinical improvement [15,16].

To classify surgical operations results with a “good early outcome”, we defined a delta of improvement of the ODI higher than 12.7/100. On the other hand, to identify surgical operations with an “excellent early outcome” we used a combined target that identifies an excellent clinical result when all three underlying targets, ODI (Δ ≥ 12.7/100), SF-36 PCS (Δ ≥ 6/100) and COMI back (Δ ≥ 2.2/10), showed a relevant improvement; good or excellent early outcomes will be defined as “Outcome +” and predicted good or excellent early outcomes will be defined as “Prediction +”. Patients with a Δ below the threshold value 12.7/100 for ODI will be considered as negative results for the good early outcome, while patients with at least one Δ value below the following thresholds, 12.7/100 for ODI, 6/100 for SF-36 PCS or 2.2/10 for COMI back, are considered negative results for the excellent early outcome; we will refer to these cases in the following as “Outcome −” or “Prediction −”. Therefore, in the analysis also patients without the minimal clinically relevant improvement as well as patients with clinical worsening were included.

An exploratory analysis was performed. Patients were classified as having or not the binomial risk factor—(Risk +) or (Risk −), respectively. Different scenarios were simulated to verify the performance of each method of calculation in three age categories. For each scenario, a 2 × 2 table was built (Good Outcome +/− vs. Risk +/−). A Chi-squared test was used for statistical association between reaching the good outcomes and the presence of risk factors. The odds-ratios with their 95% confidence intervals, and point estimations of the sensitivity and specificity of the alignment rules to discriminate patients with final good or poor clinical outcome, the positive (PPV) and negative (NPV) predictive values and positive and negative likelihood ratios (LR +, LR −, respectively) were calculated. Differences between preoperative and postoperative clinical outcomes were tested with the two-tailed Student’s t test for paired samples. The Mann–Whitney U-test was used in the cases of abnormally distributed variables. Normality was verified with the Kolmogorov–Smirnoff test. The threshold for statistical significance was set at *p* < 0.05 in all of the tests.

### 2.3. Machine Learning Approach

The available dataset is characterized by the non-negligible presence of missing values. Therefore, data imputation of independent variables was first performed to exploit all of the instances and obtain more stable and reliable results [17]. In the dataset, different data types are available and each of them was treated with dedicated techniques. In the case of numerical variables, for each instance with at least one missing value a small subset of complete instances similar to the instance under investigation was selected. From this set, a randomly sampled instance was used to replace the missing values. Discrete variables were instead imputed with ad hoc models: Logistic regression models for binary variables, multinomial logistic regression models for unordered categorical variables and ordered logit models (or proportional odds models) for categorical variables with ordered categories. The data imputation was implemented with the “mice” R package [18].

A multivariate classification model was used to predict the target variables: (1) Single ODI improvement or (2) combined ODI + SF36 PCS + COMI Back. For both targets, we used a random forest (RF) classification method to predict the outcome of the surgical operation. RF is an ensemble model composed of multiple decision trees, each of them trained independently on randomly sampled subsets of variables. The single outputs of the multiple decision tree models are then combined with a majority vote to obtain the final decision of RF. This ensemble helps in improving the predictive performance of the individual decision tree models. Indeed, RF has been recognized as one of the best performing classifiers in extensive classification studies [19], and the R implementation provided by the “randomForest” package is empirically more accurate than other implementations [20]. We thus train a RF model using the default settings of the “randomForest” R package for both targets. To evaluate the most important features used by RF to classify instances, we used the mean decrease Gini index, which measures the contribution of each variable to the homogeneity of internal and leaf nodes of the tree.

We trained RF in cross-validation (five folds) and selected the classification threshold for each fold by optimizing the geometric mean of sensitivity and specificity in a nested cross-validation loop. The proposed nested cross-validation allows to robustly estimate the optimal classification threshold and assess RF performance, while balancing sensitivity and specificity. This is particularly relevant since both the target variables are slightly unbalanced (70.7% of the available data are associated with a good surgical outcome and 43.3% of the available data are associated with an excellent surgical outcome).

The model for predicting ODI (good early outcome) at 6 months’ follow up (FU) makes use of the following features: Classification of the patient’s clinical state (Glassman), equipe operating (Equipe), age, gender, body mass index (BMI), ASA code (ASA), pre-operative medical PROMs (ODI, COMI, SF-36 Physical and SF-36 Mental), number of vertebrae stabilized during the operation (Levels), start and end points of the stabilized vertebrae (From_Level, To_Level) and comorbidity Charlson Index (Charlson).

The performance of the model is evaluated in terms of sensitivity, i.e., true positive rate (TPR), specificity, i.e., true negative rate (TNR), accuracy and area under the receiver operating characteristic curve (AUC ROC). The exploratory and further machine learning analysis were performed in R [21].

The entire study was performed according to the TRIPOD guideline for the development of multivariate models for individual prognosis or diagnosis [22].

## 3. Results

A total of 1243 patients who underwent lumbar arthrodesis surgery were included in this study. Out of them, 9.5% were disc pathologies, 38.4% were disc collapse, 32.8% were spondylolysis or spondylolisthesis, 7.6% were degenerative scoliosis, 0.1% were facet pathologies, 11.1% were non-union, 0.3% were cancer and 0.2% were infection. The rate of early good outcome was 70.7% (n = 879). A total of 43.3% (n = 538) of patients reached an “excellent” early outcome. The patients had a median age of 56 (interquartile range: 22) years and 771 (62.0%) were female. The mean baseline disability of the study population was ODI 47.3 ± 17.1, the mean pain score was COMI back 7.7 ± 1.7 and the mean quality of life was SF-36 PCS 32.7 ± 6.9 and SF-36 MCS 45.5 ± 11.8.

Since univariate exploratory analysis did not collect significant results, multivariate classification models were used to identify both surgical operations with good and excellent early outcome; the results of these analyses are reported in the Appendix A.

### Machine Learning Model

The model showed a good balance between sensitivity (74.3%) and specificity (79.4%), while providing a good accuracy (75.8%) with ROC AUC = 0.842.

This combined target model (excellent early outcome) makes use of the same features used for the good early outcome model. The excellent early outcome model showed a sensitivity of 74.2% and specificity of 71.8%, with an accuracy of 72.8%, and a ROC AUC = 0.808. Furthermore, both models for predicting good and excellent clinical outcomes showed a good balance between sensitivity (74.3% for good and 74.2% for excellent outcomes) and specificity (79.4% for good and 71.8% for excellent outcomes), while providing a good accuracy (75.8% for good and 72.8% for excellent outcomes). Details are reported in Table 2 and Table 3.

The models also showed a good discriminatory capacity of the two classes (ROC AUC = 0.842 for good and ROC AUC = 0.808 for excellent outcome). See Figure 1.

According to the mean decreasing Gini index of the random forest model, the top five predictors of both good and excellent clinical outcomes were SF-36 PCS, SF-36 MCS, ODI, BMI and age at baseline (the weights of machine learning models for good clinical outcomes were: SFMPre = 73.20, SFPPre = 70.80, ODIPre = 66.77, BMI = 62.97 and Age = 61.12; for excellent clinical outcomes they were: SFPPre = 90.34, SFMPre = 87.13, BMI = 78.61, Age = 69.92 and ODIPre = 66.21(Table 4).

The graphs of mean decreased Gini for weights of machine learning models, as well as odds ratios of the explorative analysis, are included in the Appendix A.

## 4. Discussion

In spine surgery, multiple factors can influence clinical outcomes. According to our results and the exploratory analysis, there is not a single risk factor capable of influencing or predicting early clinical outcomes. In our registry, the machine learning (ML) approach predicts the likelihood of good or excellent early clinical results. Our ML model showed good performances of post-operative prediction if based on patients’ demographic data and pre-operative self-reported degree of disability and quality of life. In the last decades, the machine learning approach in predictive models has gained interest in clinical practice.

### 4.1. Predictive Models of Surgical Improvement Based on Clinical Data

Surgical treatment for degenerative spine disorders has been shown to improve the quality of life and reduce disability in patients most severely affected [23]. Nevertheless, the associations between demographic baseline factors and overall complication rates are still unclear.

Thanks to the anesthesiological and surgical implementations, a large population can safely be a spinal surgery candidate. The increase in the use of mini-invasive surgery [24] and the improvements of pre-operative planning methods [25] has allowed enlarging the cohort of patients eligible for surgery and capable of obtaining significant results [26,27].

One of the most relevant demographic indicators is BMI (body mass index). High BMI values are known to be risk factors for many diseases and are directly correlated with the complication rate after spinal surgery, even if in the literature, BMI’s role is still debated in the prediction of functional outcomes.

According to Mulvanay et al. [28], an increased BMI is associated with decreased effectiveness of one- to three-level elective lumbar fusion, despite the absence of surgical complications. A BMI value higher than 30 is considered a risk factor for surgical complications and poor spine surgery results. According to our data, a low BMI seems to present a relevant role in predicting good clinical outcomes. Despite several studies suggesting that weight does not represent a major impact on patients’ health-related quality of life after surgery [29], obesity has a relevant impact on intraoperative blood loss, length of surgery and complication rate. It seems that BMI should always be kept in mind when planning spinal fusion. Several clinical indicators of postoperative success are continuously analyzed to improve the surgical outcome in terms of complication rate and patients’ satisfaction. The aging of the population and the relative increase of the comorbidities can challenge the surgical decision.

According to Daniels et al. [30], the upgrading of some surgical methods increased the performances of therapeutic strategies. In particular, in a retrospective analysis of surgical cases enrolled between 2009 and 2016, the complication rates decreased over time, despite an increasingly elderly, medically compromised, and obese patient population. As a critical point, the authors identified the evolution of surgical strategies that resulted in an overall improvement of the treatment quality.

### 4.2. Predictive Models of Surgical Improvement Based on PROMs

The proper estimation of the pre-operative degree of disability and quality of life is mandatory when surgery is required. In spine surgery, considerable controversy exists regarding spinal arthrodesis’ risk–benefit ratio where surgery itself creates a permanent fusion of vertebral bodies. Nevertheless, several studies demonstrated a significant improvement after spinal arthrodesis in cases of degenerative spinal disorders [26,31,32]. A combination of scales is often used in clinical studies to assess multiple aspects of human health.

The indicators of quality of life and disability progressively gained attention, becoming the gold standard to measure the success rate after spine surgery. The post-operative clinical improvement can be evaluated based on patients’ reported outcomes such as ODI. Although post-operative improvement may be statistically significant, it is not necessarily clinically relevant. For this reason, several studies have defined the values used to indicate a difference that is clinically meaningful to the patient (MCID) [33]. In particular, Monticone et al. defined as significant a cut-off value of MCID at a 12.7 ODI unit score of improvement [14].

### 4.3. Predictive Models’ Performances

Predictive models for patient-reported outcomes can improve the surgical strategies when deciding to opt for surgery or not or potentially to adapt the surgical approach.

Despite the significant variability in the population affected by a common clinical condition, lumbar disc herniation, Staartjes et al. [34] proposed a predictive model based on deep learning-based analytics. Out of a population of 422 patients, the deep learning and logistic regression attained AUC values of 0.84 and 0.72, and accuracies of 75% and 59%, respectively. The greatest discrepancy in performance measures regarded the models predicting back pain improvement. This could reflect the model’s weakness or the inherent difficulty of the outcome to be measured.

Models based on naïve Bayes machine learning to predict hospitalization days and indications for discharge (for example, admission to rehabilitation facilities or back to home and hospitalization costs) showed high performances. In particular, the system proposed by Karnuta et al. revealed a predictive accuracy of 0.800 for costs of recovery, 0.874 for length of stay (LOS), and 0.878 for disposition with AUC for hospitalization costs (0.880), an excellent AUC for LOS (0.941), and an excellent AUC for discharge disposition (0.906) [35]. The disease variability, combined with the psychological influencing factors and patients’ expectations related to the surgeries, challenges the accuracy of clinical predictions. Siccoli et al. [36] evaluated the feasibility of short- and long-term PROMs and reoperation rate using an ML approach in patients affected by lumbar stenosis. According to this study, the models were able to predict the endpoints, providing accurate information. Despite a progressive increase in the use of prediction models in spine surgery, little is known in spinal arthrodesis for two to four levels surgery.

Our results seem to provide comparable or higher predictive performances than other studies on spine surgery. Thanks to the recent advances in technologies, AI can involve the application of mathematical algorithms that continuously learn and make observations from existing data. The aim is to create a more accurate predictive model based on these data [37].

### 4.4. Influence of ML Predictions on Therapeutic Strategy

With the widening of the modern dataset, the use of ML will progressively become the gold standard and the primary candidate for the data analysis. Future application in diagnosis, prognosis and decision-making processes is desirable and will soon become an essential spine physician tool. Khan et al. introduced the application in the clinal management of cervical myelopathy and nontraumatic spinal cord injury to predict the risk of neurological impairment at one year [38]. These tools allowed the physicians to predict individual patient outcome after surgery for degenerative cervical myelopathy [39] and to apply preventive strategies such as targeted physiotherapy and the timing of psychological counselling. With the application of ML techniques, several studies demonstrated the possibility to predict clinical outcomes. Ames et al. predicted patients’ responses to SRS-22R (questionnaire) item per item up to 86.9% AUROC at 1 and 2 years following surgical treatment for ASD. The main clinical application is to aid surgical decision making during preoperative counselling [40]. In complex surgery, this approach will be capable of implementing already available surgical decision making [41].

### 4.5. Methodological Consideration and Limitations

The result of our study comes with several limitations that we have to take into account. First, the term follow-up indicators for 6 months can be considered only a preliminary result. External and prospective validations are necessary to support this methodology further so as to improve the knowledge acquired. Furthermore, the lower performance in terms of PPV in “excellent outcomes predictions” and NPV in “good outcomes predictions” can be explained by low numbers of positive and negative events, respectively. Indeed, the PPV and NPV values for the two problems show that both models perform better on the majority class (positive for good clinical outcome and negative for excellent clinical outcome). This is especially true for the “good outcome prediction”, where the imbalance is higher. In the “excellent outcome prediction”, the PPV, although lower than NPV, still highlights the model’s ability to predict the positive class. These results highlight that some predictions, i.e., negative good clinical outcome and positive excellent clinical outcome, are more difficult than the opposites, i.e., positive good clinical outcome and negative excellent clinical outcome. The predictive ability can be improved by combining the two models to obtain a classification of patients in three categories: “Excellent”, “Good” and “Not good”. This classification of patients can be used to support clinicians in making personalized and patient-specific decisions.

Although a significant role of different surgical approaches has not been identified, further studies are needed to clarify the role of different surgical techniques on medium- and long-term clinical outcomes.

## 5. Conclusions

The results of our study suggest that a machine learning approach showed high performance in predicting early good to excellent clinical results. In particular, our data suggest that with a worse score of preoperative indicators of disability and quality of life, younger or healthier patients should expect a significant clinically relevant improvement. On the other hand, older patients and patients with higher BMI, comorbidities (higher ASA and Charlson score) with higher SF-36 scores and lower ODI scores would experience less clinically relevant improvements by following the path of lumbar spine surgery. These results must be seen in light of the study’s limitations—first, the mid-term follow-up indicators, six months. A potential improvement or worsening in the PROMS results could occur later. The latter was not the focus of the study.

## Figures and Tables

**Figure 1 jpm-11-01377-f001:**
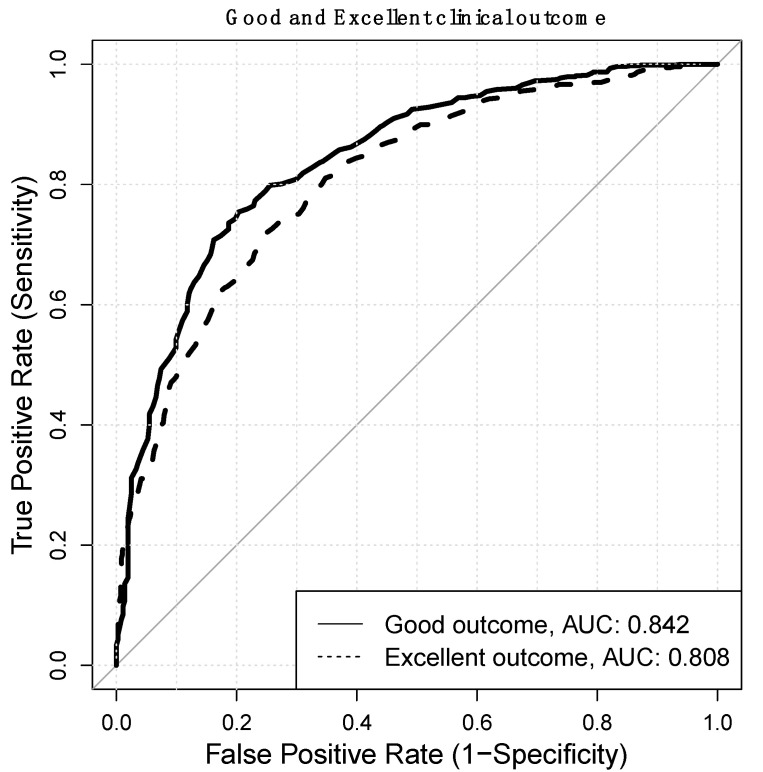
ROC CURVE for the ODI model and the combined model.

**Table 1 jpm-11-01377-t001:** Table with the features included in the analysis and associated percentage of missing values.

Glassman	Equipe	Age	Gender	BMI	ASA	ODIPre
18.8%	0.0%	0.0%	0.0%	1.0%	0.0%	0.0%
**COMIPre**	**SFPPre**	**SFMPre**	**Levels**	**From Level**	**To Level**	**CCI**
0.0%	0.0%	0.0%	20.3%	20.4%	20.4%	19.0%

Glassman: Glassman classification data; Equipe: surgical team; BMI: body mass index; ODIPre: pre-operative Oswestry disability index; COMIPre: pre-operative core outcome measures index; SFPPre: pre-operative physical component score of the short form-36; SFMPre: pre-operative mental component score of the short form-36; CCI: Charlson comorbidity index.

**Table 2 jpm-11-01377-t002:** Good clinical outcome AI predictions.

	Outcome +	Outcome −	Total		
Prediction +	653	75	728	Sensitivity	74.3%
Prediction −	226	289	515	Specificity	79.4%
Total	879	364	1243	PPV	89.7%
				NPV	56.1%
				Accuracy	75.8%
				AUC ROC	0.842

The table shows the performance evaluation of the AI model predicting the “good clinical outcome”—ODI at 6 months FU. Outcome +: patients with ODI Δ ≥ 12.7/100; Outcome −: patients with ODI Δ < 12.7/100; Prediction +: model’s predictions of patients with ODI Δ ≥ 12.7/100; Outcome −: model’s predictions of patients with ODI Δ < 12.7/100; PPV: Positive Predictive Values; NPV: Negative Predictive Values.

**Table 3 jpm-11-01377-t003:** Excellent clinical outcome AI predictions.

	Outcome +	Outcome −	Total		
Prediction +	399	199	598	Sensitivity	74.2%
Prediction −	139	506	645	Specificity	71.8%
Total	538	705	1243	PPV	66.7%
				NPV	78.4%
				Accuracy	72.8%
				AUC ROC	0.808

The table shows the performance evaluation of the AI model predicting the “Excellent Clinical Outcome”—ODI—SF36—COMI Back at 6 months FU. Outcome +: patients with ODI Δ ≥ 12.7/100 and SF-36 PCS (Δ ≥ 6/100) and COMI back (Δ ≥ 2.2/10); Outcome −: patients with at least one of the following conditions ODI Δ < 12.7/100 or SF-36 PCS Δ < 6/100 or COMI back Δ < 2.2/10; Prediction +: model’s predictions of patients with ODI Δ ≥ 12.7/100 and SF-36 PCS Δ ≥ 6/100 and COMI back Δ ≥ 2.2/10; Outcome −: model’s predictions of patients with at least one of the following conditions—ODI Δ < 12.7/100 or SF-36 PCS Δ < 6/100 or COMI back Δ < 2.2/10; PPV: Positive Predictive Values; NPV: Negative Predictive Values

**Table 4 jpm-11-01377-t004:** Mean decreased Gini weights in good and excellent clinical outcomes.

Predictive Variables	Good Clinical Outcome	Predictive Variables	Excellent Clinical Outcome
SFMPre	73.20	SFPPre	90.34
SFPPre	70.80	SFMPre	87.13
ODIPre	66.77	BMI	78.61
BMI	62.97	Age	69.92
Age	61.12	ODIPre	66.21
COMIPre	31.00	COMIPre	43.74
Glassman	28.30	Glassman	29.90

Glassman: Glassman classification data; BMI: body mass index; ODIPre: pre-operative Oswestry disability index; COMIPre: pre-operative core outcome measures index; SFPPre: pre-operative physical component score of the short form-36; SFMPre: pre-operative mental component score of the short form-36.

## Data Availability

The datasets used and/or analyzed in the present study are available from the corresponding author upon reasonable request.

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
