# Peer review of "The Influence of Baseline Clinical Status and Surgical Strategy on Early Good to Excellent Result in Spinal Lumbar Arthrodesis: A Machine Learning Approach"

_jpm, 2021, doi:10.3390/jpm11121377_

Round 1

Reviewer 1 Report

The authors aimed to develop a preoperative machine learning model to predict a good to excellent early clinical outcome by using baseline demographic and health-related quality of life scores. They conclude that ML approach showed high performance in predicting outcomes.

Strength: Important topic and information in light of the increasing of predictive models to improve cost-effectiveness.

The weaknesses of the study are related , as the authors state in the limitation section, to the short follow-up of clinical outcomes (only 6 months)

1.To my opinion, this paper will benefit from adding other features on the analisys like preoperative pathologies and type of arthrodesis. 

2. On table 2 and 3 the is a typing error about negative predictive value NPV and not PNV

3.  the lower performance in terms of NPV (56.1%) in “good outcomes predictions” and PPV (66.7%) in “excellent outcomes predictions” could influence the prediction? need to better explained in the discussion section

Author Response

Reviewer 1

The authors aimed to develop a preoperative machine learning model to predict a good to excellent early clinical outcome by using baseline demographic and health-related quality of life scores. They conclude that ML approach showed high performance in predicting outcomes.

Strength: Important topic and information in light of the increasing of predictive models to improve cost-effectiveness.

The weaknesses of the study are related , as the authors state in the limitation section, to the short follow-up of clinical outcomes (only 6 months)

Authors’ Response

Dear Reviewer, thank you for your work which produced several changes. We believe the manuscript has been significantly improved.

1.To my opinion, this paper will benefit from adding other features on the analisys like preoperative pathologies and type of arthrodesis. 

Authors’ Response

Dear Reviewer, thanks for your comment. On the one hand, pre-operative clinical conditions were included in the model and produced very relevant results. Indeed, Glassman's classification is the sixth top predictor for the Mean Decrease Gini in both prediction models. On the other hand, the Charlson comorbidity index was not represented among the main predictors of the model. We have added a table that underlines the central role of pre-operative diagnosis (Glassman's Classification).

The surgical approach in the models tested previously did not reach relevant results. This lack of supporting evidence does not entirely rule out the role of different surgical approaches. A large part of our Cohort underwent combined surgical approaches (anterior + posterior, Anterior only, posterior only and minimally invasive surgery), and therefore, further high-quality studies are needed to investigate this relevant aspect. We add this aspect within the limit section of the study.

Modified Text

“Although a significant role of different surgical approaches has not been identified, further studies are needed to clarify the role of different surgical techniques on medium and long-term clinical outcomes.”

  1. On table 2 and 3 the is a typing error about negative predictive value NPV and not PNV

Authors’ Response

Thanks. Modified text

  1.  the lower performance in terms of NPV (56.1%) in “good outcomes predictions” and PPV (66.7%) in “excellent outcomes predictions” could influence the prediction? need to better explained in the discussion section.

Authors’ Response

Thanks for the comment. This aspect was partly analyzed in the limits section of the study. We have further specified where this aspect derives from and whether it may affect the model's performance.

Modified Text

Line 437

Indeed, the PPV and NPV values for the two problems show that both models perform better on the majority class (positive for “good clinical outcome” and negative for “excellent clinical outcome”). This is especially true for the “good outcome prediction” where the imbalance is higher. In the “excellent outcome prediction” the PPV, although lower than NPV, still highlights the model's ability to predict the positive class. These results highlight that some predictions, i.e. negative “good clinical outcome” and positive “excellent clinical outcome”, are more difficult than the opposites, i.e. positive “good clinical outcome” and negative “excellent clinical outcome”. The predictive ability can be improved by combining the two models to obtain a classification of patients in three categories: “Excellent”, “Good” and “Not good”.

Reviewer 2 Report

Again, I see the names of the authors, which should be blinded for review. 

The manuscript deals with the important topic of identifying patients who benefit most from spine surgery. The ML algorithm applied seems to have added value in unraveling the key clinical factors involved. The authors present a well designed study with some important results. 

My main concerns with this study are structure and language. In the following, I provide detailed feedback. 

Oswestry Disability Index (ODI) l. 27 
please indicate the abbreviations used in the abstract

l. 21 please improve the order of the sentence, using a machine learning (ML) approach at the end of the sentence

l. 48, pain and disability represent the main feature in a way that is … - please change expression for this, I guess you are saying the perception of the condition is or so. This sentence should be improved in terms of wording

l. 52-55 no citation? please provide at least one citation per paragraph if not per sentence

l. 59, 60 should be made more clear, minimal improvements in only 25%? I got confused by “below up to”…

l. 73, 75 where do you cite? one time before the dot, one time after the dot. Get decided please

methods / results: you should provide an improved structure i.e. subheadings for methods (more than the two you have) and results please

l. 197 if you mention those results, you should provide the data. 

l. 201 what is in the model can be moved to the methods section

language: the manuscript is well written and mostly easy to read. Still, some language issues make it difficult to understand the point, so I suggest a native speaker revises the manuscript.

methods: you have defined good clinical outcome. You should also define the bad or negative clinical outcome even though the reader will assume its below ODI 12,6. especially since you provide tables below that say outcome + or - . When reading your methods section and you show the number of good and excellent clinical results, it seems not to add up at first, so please be precise about the negative outcomes as well, this makes it easier to follow and is more transparent

l. 240 is not instead of isn’t

l. 258 “is” not it’s

l. 233 I recommend presenting the clinical factors identified in a table. I think this is an important result and should be presented accordingly.

The discussion section covers all important aspects of the work including limitations.

To conclude, I recommend publishing this article after revision. 

Author Response

Reviewer 2

The manuscript deals with the important topic of identifying patients who benefit most from spine surgery. The ML algorithm applied seems to have added value in unraveling the key clinical factors involved. The authors present a well designed study with some important results. 

My main concerns with this study are structure and language. In the following, I provide detailed feedback.  

  1. Oswestry Disability Index (ODI) l. 27 
    please indicate the abbreviations used in the abstract

Authors’ Response

Thanks. Modified text

Modified Text

“…assessment (Oswestry Disability Index - ODI, SF-36 and COMI back)…”

  1. l. 21 please improve the order of the sentence, using a machine learning (ML) approach at the end of the sentence

Authors’ Response

Thanks. Modified text

Modified Text

The study aims to create a preoperative model from baseline demographic and health-related quality of life scores (HRQOL) to predict a good to excellent early clinical outcome using a machine learning (ML) approach.

  1. l. 48, pain and disability represent the main feature in a way that is … - please change expression for this, I guess you are saying the perception of the condition is or so. This sentence should be improved in terms of wording.

Authors’ Response

Thanks. Modified text

Modified text

“Pain and disability represent its main features, leading to a significant clinical and socio-economic impact with an increasing role in daily medical practice.”

  1. l. 52-55 no citation? please provide at least one citation per paragraph if not per sentence

Authors’ Response

Citation added.

  1. l. 59, 60 should be made more clear, minimal improvements in only 25%? I got confused by “below up to”…

Authors’ Response

Thanks for the comment. We also reviewed our old versions of the document and it was "be low up". However, the sentence is unclear and does not add significant information and can create confusion. That part of the sentence has been removed. The citation refers to the first data.(13% of complication rate).

  1. l. 73, 75 where do you cite? one time before the dot, one time after the dot. Get decided please

Authors’ Response

Increased the consistency in the whole text.

  1. methods / results: you should provide an improved structure i.e. subheadings for methods (more than the two you have) and results please

Authors’ Response

Done.

  1. l. 197 if you mention those results, you should provide the data. 

Authors’ Response

Done. We addressed this complementary data to Appendix A.

  1. l. 201 what is in the model can be moved to the methods section

Authors’ Response

Done.

  1. language: the manuscript is well written and mostly easy to read. Still, some language issues make it difficult to understand the point, so I suggest a native speaker revises the manuscript.

Authors’ Response

We have made multiple changes to the text. Nevertheless, we are confident that the English editing of the journal will improve the quality of the manuscript.

  1. methods: you have defined good clinical outcome. You should also define the bad or negative clinical outcome even though the reader will assume its below ODI 12,6. especially since you provide tables below that say outcome + or - . When reading your methods section and you show the number of good and excellent clinical results, it seems not to add up at first, so please be precise about the negative outcomes as well, this makes it easier to follow and is more transparent.

Authors’ Response

Thanks for the precious comment. We add a comment in the methods section. We have further improved the captions of the tables.

  1. l. 240 is not instead of isn’t

Authors’ Response

Done.

  1. l. 258 “is” not it’s

Authors’ Response

Done.

  1. l. 233 I recommend presenting the clinical factors identified in a table. I think this is an important result and should be presented accordingly.

Authors’ Response

Added. Furthermore, we have integrated the graphics in the supplement material.

The discussion section covers all important aspects of the work including limitations.

To conclude, I recommend publishing this article after revision.

Authors’ Response

Thanks.
